# An Information-Theoretic Analysis of Thompson Sampling for Logistic Bandits

**Amaury Gouverneur**
KTH Royal Institute of Technology
Stockholm, Sweden
`amauryg@kth.se`

**Borja Rodríguez-Gálvez**
KTH Royal Institute of Technology
Stockholm, Sweden
`borjarg@kth.se`

**Tobias J. Oechtering**
KTH Royal Institute of Technology
Stockholm, Sweden
`oech@kth.se`

**Mikael Skoglund**
KTH Royal Institute of Technology
Stockholm, Sweden
`skoglund@kth.se`

## Abstract

We study the performance of the Thompson Sampling algorithm for logistic bandit problems, where the agent receives binary rewards with probabilities determined by a logistic function $\exp(\beta\langle a, \theta\rangle)/(1 + \exp(\beta\langle a, \theta\rangle))$. We focus on the setting where the action $a$ and parameter $\theta$ lie within the $d$-dimensional unit ball with the action space encompassing the parameter space. Adopting the information-theoretic framework introduced by Russo and Van Roy (2015), we analyze the information ratio, which is defined as the ratio of the expected squared difference between the optimal and actual rewards to the mutual information between the optimal action and the reward. Improving upon previous results, we establish that the information ratio is bounded by $\frac{9}{2}d$. Notably, we obtain a regret bound in $O(d\sqrt{T \log(\beta T/d)})$ that depends only logarithmically on the parameter $\beta$.

## 1 Introduction

This paper studies the logistic bandit problem, where for $T$ time steps, an agent selects an action and receives a binary reward with probabilities determined by the logistic function $\exp(\beta\langle a, \theta\rangle)/(1 + \exp(\beta\langle a, \theta\rangle))$ with slope parameter $\beta > 0$. In this setting, both the action $a$ and the parameter vector $\theta$ lie within the $d$-dimensional unit ball. The logistic bandit applies to various scenarios, for instance, in personalized advertisement systems, where a platform suggests content to users who provide binary feedback, such as "like" or "dislike" (Chapelle and Li, 2011; Russo and Van Roy, 2017).

The performance, or regret, of algorithms for logistic bandits, has been extensively studied, with significant contributions including analyses of Upper-Confidence-Bound (UCB) algorithms by Filippi et al. (2010), Li et al. (2017) and Faury et al. (2020) as well as the study of Thompson Sampling (TS) by Russo and Van Roy (2014) and Abeille and Lazaric (2017). However, nearly all existing regret bounds exhibit an exponential dependence on the parameter $\beta$. This dependence is highly unsatisfactory because, in practice, as $\beta$ increases, it is faster to identify the optimal actions, as the distinction between near-optimal and suboptimal actions becomes more pronounced.

In this work, we focus on the TS algorithm (Thompson, 1933), which, despite its simplicity, has proven to be highly effective across a wide range of problems (Russo et al., 2018; Chapelle and Li, 2011). To analyze the regret of TS, Russo and Van Roy (2015) introduced the concept of the information ratio, a statistic that quantifies the trade-off between the information gained about

Workshop on Bayesian Decision-making and Uncertainty, 38th Conference on Neural Information Processing Systems (NeurIPS 2024).

the parameter and the immediate regret incurred. Dong and Van Roy (2018) conjectured that the information ratio for TS in logistic bandits could be bounded solely by the problem's dimension $d$, and several studies have since aimed to characterize this ratio for logistic bandits.

Recently, Neu et al. (2022) derived a regret bound of $O(\sqrt{dT|\mathcal{A}|\log(\beta T)})$ on the logistic bandit problem, but their result relies on a worst case TS information ratio bound scaling with the cardinality of the action space $|\mathcal{A}|$. Dong et al. (2019) provided a bound of $100d$ on the information ratio for TS when $\beta < 2$. They also suggested, through numerical computations, that this bound holds for larger values of $\beta$. However, their work has two key limitations. First, they do not provide a rigorous proof for generalizing to larger $\beta$ values. Second, and more critically, their regret analysis is incomplete as it relies on the rate-distortion bound from Dong and Van Roy (2018), which specifically requires a bound on the *one-step compressed TS* information ratio; a fundamentally different quantity from the TS information ratio they studied. Notably, their techniques to bound the TS information ratio do not apply to the one-step compressed TS information ratio.

In this work, we address these issues and propose a regret bound that scales only logarithmically with the slope of the logistic function. Our key contributions are as follows:

- We propose an information-theoretic regret bound for infinite and continuous action and parameter spaces that relies on the entropy of the quantized parameter, $H(\Theta_\varepsilon)$. This result is achieved by adapting the approaches from Neu et al. (2021) and Gouverneur et al. (2023).

- We present a refined analysis showing that for all $\beta > 0$, the information ratio of TS for logistic bandits is bounded by $\frac{9}{2}d$, improving upon previous results.

- We establish a bound of $O(d\sqrt{T\log(\beta T/d)})$ on the regret of TS. To our knowledge, this is the first bound on logistic bandits that scales only logarithmically on $\beta > 0$ and is independent of the number of actions.

## 2 Problem Setup

We consider a logistic bandit problem, where at each time step $t \in \{1, \ldots, T\}$, an agent selects an action $A_t$ and receives a binary reward $R_t \in \{0, 1\}$ with probability following a logistic function:

$$\mathbb{P}(R_t = 1 | A_t = a, \Theta = \theta) = \frac{\exp(\beta\langle a, \theta\rangle)}{1 + \exp(\beta\langle a, \theta\rangle)}.$$

Here, $\beta$ is a known scale parameter, and $\langle a, \theta\rangle$ denotes the inner product of the action vector $a \in \mathcal{A}$ and the unknown parameter $\theta \in \mathcal{O}$. The logistic function, sometimes referred to as the link function, is denoted $\phi_\beta(\langle a, \theta\rangle)$. In this setting, the action $a$ lies within the $d$-dimensional Euclidean unit ball, $\mathbf{B}_d(0, 1)$, and the parameter vector $\theta$ on the $d$-dimensional Euclidean unit sphere, $\mathbf{S}_d(0, 1)$. Additionally, we assume the action space $\mathcal{A}$ encompasses the parameter space $\mathcal{O}$, that is $\mathcal{O} \subseteq \mathcal{A}$. Note that this ensures that, for each $\theta \in \mathcal{O}$, there exist an action $a \in \mathcal{A}$ equal to $\theta$, such that $\langle a, \theta\rangle = 1$.

Following the Bayesian framework, we assume the parameter vector $\theta$ is sampled from a known prior distribution $\mathbb{P}_\Theta$. As the reward distribution depends only on the selected action and the parameter, it can be written as $R_t = R(A_t, \Theta)$. The agent's history at time $t$ is denoted by $H^t = \{A_1, R_1, \ldots, A_{t-1}, R_{t-1}\}$, representing all past actions and rewards observed up to time $t$.

The goal of the agent is to sequentially select actions that maximize the total accumulated reward, or equivalently, that minimize the total expected regret defined as:

$$\mathbb{E}[\text{Regret}(T)] := \mathbb{E}\left[\sum_{t=1}^{T} R(A^\star, \Theta) - R(A_t, \Theta)\right],$$

where $A^\star$ is the *optimal action* for the parameter $\Theta$. We construct the optimal mapping $\pi^\star(\theta) := \text{argmax}_{a \in \mathcal{A}}\mathbb{E}[R(a, \theta)]$ and define $A^\star = \pi^\star(\Theta)$. The expectation is taken over the randomness of the action selection, the reward distribution, and the prior distribution of the parameter $\Theta$.

Since the $\sigma$-algebras of the history are often used in conditioning, we introduce the notations $\mathbb{E}_t[\cdot] := \mathbb{E}[\cdot | H^t]$ and $\mathbb{P}_t[\cdot] := \mathbb{P}[\cdot | H^t]$ to denote the conditional expectation and probability given $H^t$. Additionally, we define $I_t(A^\star; R_t | A_t) := \mathbb{E}_t[D_{KL}(\mathbb{P}_{R_t|H^t,A^\star,A_t} \| \mathbb{P}_{R_t|H^t,A_t})]$ as the disintegrated conditional mutual information between the optimal action $A^\star$ and the reward $R_t$ conditioned on the action $A_t$, *given the history* $H^t$.

# 3 Thompson Sampling, Information ratio, and Quantization

An elegant algorithm for solving bandit problems is the *Thompson Sampling* algorithm. It works by randomly selecting actions according to their posterior probability of being optimal. More specifically, at each time step $t \in \{1, \ldots, T\}$, the agent samples a parameter estimate $\hat{\Theta}_t$ from the posterior distribution conditioned on the history $H^t$ and selects the action that is optimal for the sampled parameter estimate, $A_t = \pi^\star(\hat{\Theta}_t)$. The pseudocode for TS is given in Algorithm 1.

---
**Algorithm 1** Thompson Sampling algorithm

---
1: **Input:** parameter prior $\mathbb{P}_\Theta$.
2: **for** $t = 1$ **to** T **do**
3:     Sample a parameter estimate $\hat{\Theta}_t \sim \mathbb{P}_{\Theta|H^t}$.
4:     Take the corresponding optimal action $A_t = \pi^\star(\hat{\Theta}_t)$.
5:     Collect the reward $R_t = R(A_t, \Theta)$.
6:     Update the history $H^{t+1} = H^t \cup \{A_t, R_t\}$.
7: **end for**

---

Studying the regret of TS in bandit problems, Russo and Van Roy (2015) introduced a key quantity to the analysis, the *information ratio* defined as the following random variable:

$$\Gamma_t := \frac{\mathbb{E}_t[R(A^\star, \Theta) - R(A_t, \Theta)]^2}{\mathrm{I}_t(A^\star; R(A_t, \Theta), A_t)}.$$

This ratio measures the trade-off between minimizing the current squared regret and gathering information about the optimal action. Russo and Van Roy use this concept to provide a general regret bound that depends on the time horizon $T$, the entropy of the prior distribution of $A^\star$, and an algorithm- and problem-dependent upper bound $\Gamma$ on the average expected information ratio (Russo and Van Roy, 2015, Proposition 1).

A limitation of this approach is that the prior entropy of the optimal action, $\mathrm{H}(A^\star)$, can grow arbitrarily large with the number of actions or get infinite if the action space is continuous. We address this issue with Theorem 1, where we propose a regret bound that depends instead on the entropy of $\Theta_\varepsilon$, a quantized version of the parameter $\Theta$. The quantized parameter, defined in Definition 1, is obtained by setting $\Theta_\varepsilon$ as the closest approximation for $\Theta$ on an $\varepsilon$-net for the metric space $(\mathcal{O}, \rho)$.

**Definition 1** *Let the set $\mathcal{O}_\varepsilon$ be an $\varepsilon$-net for $(\mathcal{O}, \rho)$ with associated projection mapping $q : \mathcal{O} \to \mathcal{O}_\varepsilon$ such that for all $\theta \in \mathcal{O}$ we have $\rho(\theta, q(\theta)) \leq \varepsilon$. We define the* quantized parameter *as $\Theta_\varepsilon := q(\Theta)$.*

# 4 Main Results

This section presents our main results on the regret of TS for logistic bandits. In Theorem 1, we start by leveraging the previously introduced concepts to derive an information-theoretic regret bound that holds for continuous and infinite parameter spaces. Following this, we state in Proposition 1 our principal contribution, where we prove a bound of $\frac{9}{2}d$ on the TS information ratio. Combining this result with our regret bound, we derive in Theorem 2, a bound on the expected regret of TS for logistic bandits, which scales as $O(d\sqrt{T \log(\beta T/d)})$.

The first theorem we present is an adaptation of (Gouverneur et al., 2023, Theorem 2) and (Neu et al., 2021, Theorem 2) to the logistic bandits setting. It relates the regret of TS to the entropy of the quantized parameter $\Theta_\varepsilon$.

**Theorem 1** *Under the logistic bandit setting with logistic function $\phi_\beta(x) = e^{\beta x}/(1 + e^{\beta x})$, let $\Theta_\varepsilon$ be defined as in Definition 1 for some $\varepsilon > 0$. Assume that the average expected TS information ratio is bounded, $\frac{1}{T} \sum_{t=1}^T \mathbb{E}[\Gamma_t] \leq \Gamma$, for some $\Gamma > 0$. Then, the TS cumulative regret is bounded as*

$$\mathbb{E}[\mathrm{Regret}(T)] \leq \sqrt{\Gamma T \left(\mathrm{H}(\Theta_\varepsilon) + \varepsilon \beta T\right)}.$$

Notably, the above theorem holds for continuous action and parameter spaces and works with bounds on the average expected information ratio of the *"standard"* TS, instead of the one-step compressed

TS as in (Dong and Van Roy, 2018, Theorem 1). This distinction is crucial for effectively controlling the information ratio. We explore this difference in more detail in Appendix B.

The proof of Theorem 1 relies on an approximation of the conditional mutual information $I(\Theta; R_t|A_t, H^t)$ as $I(\Theta_\varepsilon; R_t|A_t, H^t) + \beta\varepsilon$ exploiting the fact that, for all $a \in \mathcal{A}$ and $\theta \in \mathcal{O}$, the log-likelihood of $R(a, \theta)$ is $\beta$-Lipschitz with respect to $\theta$. The proof is presented in Appendix A.

In the following proposition, we present our main contribution, a bound on the information ratio of TS that depends only on the dimension $d$ of the problem. This result confirms, under the considered setting, and up to a multiplicative factor of 9, the conjecture of Dong and Van Roy (2018).

**Proposition 1** *Under the logistic bandit setting with logistic function $\phi_\beta(x) = e^{\beta x}/(1 + e^{\beta x})$, let $\mathcal{A} \subseteq \mathbf{B}_d(0, 1)$ and $\mathcal{O} \subseteq \mathbf{S}_d(0, 1)$ be such that $\mathcal{O} \subseteq \mathcal{A}$. Then, for all $\beta > 0$, the TS information ratio is bounded as $\Gamma_t \leq \frac{9}{2}d$.*

The proof of Proposition 1 is presented in Appendix B. At a high level, our proof consists of three parts: a lower bound on the mutual information, an upper bound on the squared expected regret at time $t$, and an upper bound on a ratio of expected variances by the study of the limit case $\beta \to \infty$. A quantity that plays a central role in our analysis is the expected variance of the reward probability $\phi_\beta(\langle A_t, \Theta\rangle)$ conditioned on $\Theta$, $\mathbb{E}_t[\mathbb{V}_t[\phi_\beta(\langle A_t, \Theta\rangle)|\Theta]]$. It is used as a lower bound on the mutual information and a related quantity shows up in the upper bound on the squared expected regret.

By combining Proposition 1 with Theorem 1, we arrive at our main result: a bound on the expected TS regret that scales as $O(d\sqrt{T\log(\beta T/d)})$.

**Theorem 2** *Under the logistic bandit setting with logistic function $\phi_\beta(x) = e^{\beta x}/(1 + e^{\beta x})$, let $\mathcal{A} \subseteq \mathbf{B}_d(0, 1)$ and $\mathcal{O} \subseteq \mathbf{S}_d(0, 1)$ be such that $\mathcal{O} \subseteq \mathcal{A}$. Then for all $\beta > 0$, the TS regret is bounded as*

$$\mathbb{E}[\text{Regret}(T)] \leq 3d\sqrt{T\log\left(\sqrt{3 + \frac{6\beta T}{d}}\right)}.$$

**Sketch of proof** *After combining Theorem 1 with Proposition 1, we upper bound the entropy $\mathrm{H}(\Theta_\varepsilon)$ by the cardinality of the $\varepsilon$-net to get a regret bound of $3\sqrt{d/2T\left(\log(|\Theta_\varepsilon|) + \varepsilon\beta T\right)}$. As the parameter space $\mathcal{O}$ is within the Euclidean unit ball, we can use Lemma 8 to control the covering number as $\log(|\Theta_\varepsilon|) \leq d\log(1 + 2/\varepsilon)$. Finally, setting $\varepsilon = d/(\beta T)$ and rearranging terms inside the logarithm yields the desired result.*

To the best of our knowledge, this is the first regret bound for logistic bandits that scales only logarithmically with the logistic function's parameter $\beta$ while remaining independent of the number of actions. We note that it is within a factor of $O(\sqrt{\log(\beta T/d)})$ of the minimax lower bound $\Omega(d\sqrt{T})$ from (Dani et al., 2008).

## 5 Conclusion and Future Work

In this paper, we studied the Bayesian regret of the Thompson Sampling algorithm for sequential decision-making problems under uncertainty, focusing on logistic bandit problems with action and parameter spaces in the $d$-dimensional unit ball. We improved the state-of-the-art bounds, proving that when the action space $\mathcal{A}$ encompasses the parameter space $\mathcal{O}$, the information ratio of TS is bounded by $\frac{9}{2}d$. Using this result, we established that TS expected regret is bounded by $O(d\sqrt{T\log(\beta T/d)})$.

A natural direction for future work is to extend our bounds to settings where the action space does not fully encompass the parameter space. This extension requires careful analysis of how well the action space aligns with the parameter space, a property closely related to the *fragility dimension*, $\eta(\mathcal{A}, \mathcal{O})$, introduced by Dong et al. (2019). This quantity is crucial for the analysis of logistic bandits, as Dong et al. (2019) demonstrated that there cannot be an $\eta(\mathcal{A}, \mathcal{O})$-independent upper bound that is both polynomial in $d$ and sub-linear in $T$. In cases where the action space does encompass the parameter space, this fragility dimension is minimal, equal to $d + 1$. However, in problems where this relation is not satisfied and with dimension $d \geq 3$, the fragility dimension can grow significantly and become as large as the cardinality of the action set. Future research should take this challenge into consideration to develop regret bounds applicable to more general settings.

**Acknowledgments**

We would like to thank Benjamin Van Roy and Yifan Zhu for the insightful conversations.

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

# A Proof of Theorem 1

**Theorem 1** *Under the logistic bandit setting with logistic function $\phi_\beta(x) = e^{\beta x}/(1 + e^{\beta x})$, let $\Theta_\varepsilon$ be defined as in Definition 1 for some $\varepsilon > 0$. Assume that the average expected TS information ratio is bounded, $\frac{1}{T}\sum_{t=1}^{T}\mathbb{E}[\Gamma_t] \leq \Gamma$, for some $\Gamma > 0$. Then, the TS cumulative regret is bounded as*

$$\mathbb{E}[\mathrm{Regret}(T)] \leq \sqrt{\Gamma T\left(\mathrm{H}(\Theta_\varepsilon) + \varepsilon\beta T\right)}.$$

**Proof 1** *We start by rewriting the expected regret of TS using the information ratio:*

$$\mathbb{E}[\mathrm{Regret}(T)] = \sum_{t=1}^{T}\mathbb{E}[R(A^\star, \Theta) - R(A_t, \Theta)]$$

$$= \sum_{t=1}^{T}\mathbb{E}\left[\sqrt{\Gamma_t\mathrm{I}_t(A^\star; R(A_t, \Theta), A_t)}\right].$$

*We continue using Jensen's inequality and applying Cauchy-Schwartz inequality:*

$$\mathbb{E}[\mathrm{Regret}(T)] \leq \sum_{t=1}^{T}\sqrt{\mathbb{E}[\Gamma_t]\mathrm{I}(A^\star; R(A_t, \Theta), A_t|H^t)}$$

$$\leq \sqrt{\Gamma T\sum_{t=1}^{T}\mathrm{I}(A^\star; R(A_t, \Theta), A_t|H^t)}$$

*where in the last inequality, we used that $\sum_{t=1}^{T}\mathbb{E}_t[\Gamma_t] \leq \Gamma T$. Applying the chain rule (Yury Polyanskiy, 2022, Theorem 3.7.b) we can decompose the mutual information as*

$$\mathrm{I}(A^\star; R(A_t, \Theta), A_t|H^t) = \mathrm{I}(A^\star; A_t|H^t) + \mathrm{I}_t(A^\star; R(A_t, \Theta)|H^t, A_t)$$

$$= \mathrm{I}(A^\star; R(A_t, \Theta)|H^t, A_t),$$

*where we used the fact that the mutual information $\mathrm{I}_t(A^\star; A_t|H^t) = 0$ as the optimal action $A^\star$ and the TS action $A_t$ are independent conditioned on the history $H^t$.*

*Let $f_{R_t|H^t, A_t, \Theta}$ denote the probability density of $R_t$ conditioned on $H^t, A_t, \Theta$ and $f_{R_t|H^t, A_t}$ denote the probability density on $H^t, A_t$. Then, the mutual information terms can be written as*

$$\mathrm{I}(\Theta; R_t|H^t, A_t) = \mathbb{E}\left[\log\frac{f_{R_t|H^t, A_t, \Theta}(R_t)}{f_{R_t|H^t, A_t}(R_t)}\right].$$

*We let the set $\mathcal{O}_\varepsilon$ be an $\varepsilon$-net for $(\mathcal{O}, \rho)$ with associated projection mapping $q : \mathcal{O} \to \mathcal{O}_\varepsilon$ such that for all $\theta \in \mathcal{O}$ we have $\rho(\theta, q(\theta)) \leq \varepsilon$. Similarly, as in the proof of (Neu et al., 2021, Theorem 2), we note that the mutual information can equivalently be written as*

$$\mathbb{E}\left[\int_{\mathcal{O}}f_{\Theta|R_t, H^t, A_t}(\theta)\left(\log\frac{f_{R_t|A_t, \Theta=\theta}(R_t)}{f_{R_t|A_t, \Theta=q(\theta)}(R_t)} + \log\frac{f_{R_t|H^t, A_t, \Theta=q(\theta)}(R_t)}{f_{R_t|H^t, A_t}(R_t)}\right)d\theta\right], \quad (1)$$

*since $f_{R_t|H^t, A_t, \Theta} = f_{R_t|A_t, \Theta}$ a.s. by the conditional Markov chain $R_t - A_t - H_t \mid \Theta$.*

*Since the derivative of $\log(\phi_\beta(x))$ is equal to $\beta/(1 + \exp(\beta x))$, it is bounded by $\beta$ and is therefore $\beta$-Lipschitz. As for all $a \in \mathcal{A}$ and all $\theta \in \mathcal{O}$, the inner product $\langle a, \theta\rangle \leq 1$, we conclude that $\log(f_{R_t|A_t, \Theta=\theta}(1))$ is $\beta$-Lipschitz with respect to $\theta$. Similarly $\frac{d}{dx}\log(1-\phi_\beta(x)) = -\beta\exp(\beta x)/(1+\exp(\beta x))$ is also bounded by $\beta$ and is therefore $\beta$-Lipschitz. We get that $\log(f_{R_t|A_t, \Theta=\theta}(0))$ is $\beta$-Lipschitz with respect to $\theta$. We conclude that $|\log f_{R_t|A_t, \Theta=\theta}(R_t) - \log f_{R_t|A_t, \Theta=q(\theta)}(R_t)| \leq \beta\rho(\theta, q(\theta)) \leq \beta\varepsilon$.*

*Then, defining the random variable $\Theta_\varepsilon := q(\Theta)$, we note that the second term in (1) is equal to $\mathrm{I}(\Theta_\varepsilon; R_t|H^t, A_t)$. Summing the $T$ mutual information $\mathrm{I}(\Theta_\varepsilon; R_t|H^t, A_t)$ and applying the chain rule (see (Yury Polyanskiy, 2022, Theorem 3.7.e)), we obtain*

$$\mathbb{E}[\mathrm{Regret}(T)] \leq \sqrt{\Gamma T\left(\mathrm{I}(\Theta_\varepsilon; H^T) + \varepsilon\beta T\right)}.$$

*Finally, we upper bound the mutual information $\mathrm{I}(\Theta_\varepsilon; H^T)$ by the entropy $\mathrm{H}(\Theta_\varepsilon)$ as in (Yury Polyanskiy, 2022, Theorem 3.4.e) to obtain the claimed result.*

# B  Proof of Proposition 1

This section presents the key ideas of the proofs of our main contribution, Proposition 1.

**Proposition 1** *Under the logistic bandit setting with logistic function $\phi_\beta(x) = e^{\beta x}/(1 + e^{\beta x})$, let $\mathcal{A} \subseteq \mathbf{B}_d(0,1)$ and $\mathcal{O} \subseteq \mathbf{S}_d(0,1)$ be such that $\mathcal{O} \subseteq \mathcal{A}$. Then, for all $\beta > 0$, the TS information ratio is bounded as $\Gamma_t \leq \frac{9}{2}d$.*

Under the logistic bandit setting with link function $\phi_\beta$, the reward $R(A_t, \Theta)$ is given by a Bernoulli random variable with associated probability $\phi_\beta(\langle A_t, \Theta \rangle)$. We use the notation $\mathrm{Bern}(\phi_\beta(\langle A_t, \Theta \rangle))$ to make the setting more explicit. Since we assumed the action space $\mathcal{A}$ encompasses the parameter space $\mathcal{O}$ and both are subsets of a $d$-dimensional unit sphere, the optimal action is to select the action that aligns with the parameter, $\pi^\star(\theta) = \theta$. We can therefore write $R(A^\star, \Theta) = R(\pi^\star(\Theta), \Theta)$ equivalently as $\mathrm{Bern}(\phi_\beta(\langle \Theta, \Theta \rangle))$ and similarly, write $R(A_t, \Theta)$ as $\mathrm{Bern}(\phi_\beta(\langle \hat{\Theta}_t, \Theta \rangle))$. To alleviate the writing, we will omit the subscript $t$ notation for the rest of the section.

Equipped with this new notation, we recall the definition of the information ratio:

$$\Gamma(\Theta, \hat{\Theta}) = \frac{\mathbb{E}[\mathrm{Bern}(\phi_\beta(\langle \Theta, \Theta \rangle)) - \mathrm{Bern}(\phi_\beta(\langle \hat{\Theta}, \Theta \rangle))]^2}{\mathrm{I}(\Theta; \mathrm{Bern}(\phi_\beta(\langle \hat{\Theta}, \Theta \rangle)), \hat{\Theta})}.$$

Our proof can be broadly divided into three key components: establishing a lower bound on the mutual information, deriving an upper bound on the squared expected regret, and obtaining an upper bound on a ratio of expected variances by analyzing the limit case as $\beta \to \infty$. A crucial element in our analysis is the expected variance of $\phi_\beta(\langle \hat{\Theta}, \Theta \rangle)$ conditioned on $\Theta$, expressed as $\mathbb{E}[\mathbb{V}[\phi_\beta(\langle \hat{\Theta}, \Theta \rangle)|\Theta]]$. It appears in the lower bound on mutual information and a related quantity is used to upper bound the squared expected regret.

## B.1  Lower bounding the mutual information

We start by giving a general lemma that relates the mutual information between a random variable $U$ and a Bernoulli random variable with probability $U$ to the variance of the random variable $U$.

**Lemma 1** *Let $U$ be a random variable taking values in $[0,1]$ and $\mathrm{Bern}(U)$ be a Bernoulli random variable with probability $U$. Then it holds that,*

$$\mathrm{I}(U; \mathrm{Bern}(U)) \geq 2\mathbb{V}(U).$$

**Proof 2** *The proof uses the decomposition of mutual information as a difference of entropy and the Taylor expansion of the binary entropy function. Using proposition Yury Polyanskiy (2022)[Theorem 3.4.d], we can decompose the mutual information between $U$ and $\mathrm{Bern}(U)$ as*

$$\mathrm{I}(U; \mathrm{Bern}(U)) = h(\mathrm{Bern}(U)) - h(\mathrm{Bern}(U)|U)$$

*Following Duchi (2016)[Example 2.2] notation, we define $h_2(p) := -p\log(p) - (1-p)\log(1-p)$ for $p \in [0,1]$. We note that we can rewrite the mutual information as*

$$\mathrm{I}(U; \mathrm{Bern}(U)) = h_2(\mathbb{E}[U]) - \mathbb{E}[h_2(U)]. \tag{2}$$

*From a Taylor expansion of $h_2(x)$ we have that*

$$h_2(x) = h_2(p) + (x-p)h_2'(p) + \frac{1}{2}(x-p)^2 h_2''(\xi)$$

*for some $\xi \in (0,1)$. We can compute the second derivative of $h_2$ and get $h_2''(\xi) = -\frac{1}{\xi(1-\xi)}$. This function is concave and is maximal at $\xi = 1/2$, where it takes the value $h_2''(1/2) = -4$. We then have that for all $x \in [0,1]$ and all $p \in [0,1]$,*

$$h_2(x) \leq h_2(p) + (x-p)h_2'(p) - 2(x-p)^2.$$

*Using this fact for $x = U$ and $p = \mathbb{E}[U]$, we have that*

$$h_2(U) \leq h_2(\mathbb{E}[U]) + (U - \mathbb{E}[U])h_2'(\mathbb{E}[U]) - 2(U - \mathbb{E}[U])^2.$$

*Applying the last inequality to the second term in (2), it comes that*

$$\mathrm{I}(U; \mathrm{Bern}(U)) \geq \mathbb{E}\left[h_2(\mathbb{E}[U]) - h_2(\mathbb{E}[U]) - (U - \mathbb{E}[U])h_2'(\mathbb{E}[U]) + 2(U - \mathbb{E}[U])^2\right].$$

*Finally, simplifying terms and taking the expectation gives the desired result.*

Equipped with Lemma 1, we can now state and prove our lower bound on the mutual information $\mathrm{I}(\Theta; \mathrm{Bern}(\phi_\beta(\langle\hat{\Theta}, \Theta\rangle)), \hat{\Theta})$.

**Lemma 2** *Let the link function be $\phi_\beta(x) = e^{\beta x}/(1 + e^{\beta x})$, then, it holds that*

$$\mathrm{I}(\Theta; \mathrm{Bern}(\phi_\beta(\langle\hat{\Theta}, \Theta\rangle)), \hat{\Theta}) \geq 2\mathbb{E}\left[\mathbb{V}\left[\phi_\beta(\langle\hat{\Theta}, \Theta\rangle)\mid \Theta\right]\right].$$

**Proof 3** *We start the proof by applying the chain rule. It comes that*

$$\begin{aligned}
&\mathrm{I}\left(\Theta; \hat{\Theta}, \mathrm{Bern}\left(\phi_\beta(\langle\hat{\Theta}, \Theta\rangle)\right)\right) \\
&= \mathrm{I}\left(\Theta; \hat{\Theta}\right) + \mathrm{I}\left(\Theta; \mathrm{Bern}\left(\phi_\beta(\langle\hat{\Theta}, \Theta\rangle)\right) \mid \hat{\Theta}\right) \\
&\overset{(i)}{=} \mathrm{I}\left(\Theta; \mathrm{Bern}\left(\phi_\beta(\langle\hat{\Theta}, \Theta\rangle)\right) \mid \hat{\Theta}\right) \\
&\overset{(j)}{=} \mathrm{I}\left(\hat{\Theta}; \mathrm{Bern}\left(\phi_\beta(\langle\hat{\Theta}, \Theta\rangle)\right) \mid \Theta\right) \\
&\overset{(k)}{=} \mathbb{E}[\mathrm{I}\left(\phi_\beta(\langle\hat{\Theta}, \Theta\rangle)); \mathrm{Bern}\left(\phi_\beta(\langle\hat{\Theta}, \Theta\rangle)\right)\right) \mid \Theta = \theta],
\end{aligned}$$

*where (i) follows as $\Theta$ and $\hat{\Theta}$ are independent conditioned on the history; (j) follows as $\Theta$ and $\hat{\Theta}$ are identically distributed conditioned on the history; and (k) is obtained by taking the expectation over $\Theta$ and using (Yury Polyanskiy, 2022, Theorem 3.2.d) as $\phi_\beta(\langle\theta, x\rangle)$ is a one-to-one mapping. Finally, applying Lemma 1 yields the desired result.*

*We note that this result would not be possible to obtain for mutual information of the "one-step compressed Thompson Sampling" as equality (j) requires $\Theta$ and $\hat{\Theta}$ to be identically distributed.*

### B.2   Upper bounding the squared expected regret

A quantity that will naturally come up often is the expected difference between of the parameter $\Theta$ and the sampled ones $\hat{\Theta}$ and how it relates to the expected difference of rewards $\phi_\beta(\langle\Theta, \Theta\rangle) - \phi_\beta(\langle\hat{\Theta}, \Theta\rangle)$.

To alleviate the notations, we define $\psi_\beta(x) := \phi_\beta(1) - \phi_\beta(1 - x)$.

Observing that $\langle\Theta, \Theta\rangle = \|\Theta\|_2^2 = 1$, we have indeed

$$\phi_\beta(\langle\Theta, \Theta\rangle) - \phi_\beta(\langle\hat{\Theta}, \Theta\rangle) = \phi_\beta(1) - \phi_\beta(\langle\hat{\Theta}, \Theta\rangle) = \psi_\beta(1 - \langle\hat{\Theta}, \Theta\rangle).$$

The two following lemmata from Dong et al. (2019) will be of importance for our analysis.

**Lemma 3 ((Dong et al., 2019, Lemma 16))** *Let $U, V$ be random vectors in $\mathbb{R}^d$, and let $\tilde{U}, \tilde{V}$ be independent random variables with distributions equal to the marginals of $U, V$, respectively. Then*

$$\mathbb{E}\left[\left(U^\top V\right)\right]^2 \leq d \cdot \mathbb{E}\left[\left(\tilde{U}^\top \tilde{V}\right)^2\right].$$

**Lemma 4 ((Dong et al., 2019, Lemma 18))** *Let $f : \mathbb{R}_+ \to \mathbb{R}_+$ be such that $f(0) \geq 0$ and $f(\zeta)/\zeta$ is non-decreasing over $\zeta \geq 0$. Then, for any non-negative random variable $U$, there is*

$$\frac{\mathbb{E}[f(U)]^2}{\mathbb{E}[U]^2} \leq \frac{\mathbb{V}[f(U)]}{\mathbb{V}[U]}.$$

The function $\psi_\beta(x)$ satisfies the first two requirements from Lemma 4: applied on the difference of inner products $\langle\Theta, \Theta\rangle - \langle\hat{\Theta}, \Theta\rangle$, it is a mapping from $[0, 2]$ to $[0, 1]$ and $\psi_\beta(0) = \phi_\beta(1) - \phi_\beta(1 - 0) = 0$. However, it fails to satisfy the third requirement; $\psi_\beta(x)/x$ increases initially, reaching a maximum between $1$ and $2$ before decreasing (see in Remark 1). This issue leads to the introduction of a modified function, which we call the *logistic surrogate*, as the tightest upper bound $\psi_\beta(x)$ on that satisfies the last requirement from Lemma 4.

**Definition 2 (Logistic surrogate)** *We construct the* logistic surrogate *function $\varphi_\beta$ as the tightest upper bound on $\psi_\beta(x)$ such that $\varphi_\beta(x)/x$ is non-decreasing over $x \geq 0$.*

*Namely, let $\delta_\beta = \arg\max_{x \in [0,2]} \frac{\psi_\beta(x)}{x}$, we define the function $\varphi_\beta$ as*

$$\varphi_\beta(x) = \begin{cases} \psi_\beta(x) & x \in [0, \delta_\beta], \\ \psi_\beta(\delta_\beta) + (x - \delta_\beta) \cdot \psi_\beta(\delta_\beta)/\delta_\beta & x \in ]\delta_\beta, 2]. \end{cases}$$

We are now equipped to state and prove an upper bound on the squared expected regret.

**Lemma 5** *Let the logistic surrogate be defined as in Definition 2. Then, it holds that*

$$\mathbb{E}[\text{Bern}(\phi_\beta(\langle \Theta, \Theta \rangle)) - \text{Bern}(\phi_\beta(\langle \hat{\Theta}, \Theta \rangle))]^2 \leq d \cdot \mathbb{E}\left[ \mathbb{V}\left[ \varphi_\beta(1 - \langle \hat{\Theta}, \Theta \rangle) \mid \Theta \right] \right].$$

**Proof 4** *Integrating over the randomness of the Bernoulli outcome, we can write the squared expected regret as $\mathbb{E}[\phi_\beta(\langle \Theta, \Theta \rangle) - \phi_\beta(\langle \hat{\Theta}, \Theta \rangle)]^2 = \mathbb{E}[\psi_\beta(1 - \langle \hat{\Theta}, \Theta \rangle)]^2$. By the definition of $\varphi_\beta$, we have $\varphi_\beta(x) \geq \psi_\beta(x)$. Using this and the law of total expectation, we have that*

$$\mathbb{E}[\psi_\beta(1 - \langle \hat{\Theta}, \Theta \rangle)]^2 \leq \mathbb{E}[\mathbb{E}[\varphi_\beta(1 - \langle \hat{\Theta}, \Theta \rangle)|\Theta]]^2.$$

*We now apply Lemma 4 on $\mathbb{E}[\varphi_\beta(1 - \langle \hat{\Theta}, \Theta \rangle)|\Theta]$. It comes that*

$$\mathbb{E}[\mathbb{E}[\varphi_\beta(1 - \langle \hat{\Theta}, \Theta \rangle)|\Theta]]^2 \leq \mathbb{E}\left[ \underbrace{\sqrt{\frac{\mathbb{V}\left[ \varphi_\beta\left(1 - \langle \hat{\Theta}, \Theta \rangle\right) \mid \Theta \right]}{\mathbb{V}\left[ 1 - \langle \hat{\Theta}, \Theta \rangle \mid \Theta \right]}} \mathbb{E}\left[ 1 - \langle \hat{\Theta}, \Theta \rangle \mid \Theta \right]}_{:=U(\Theta)} \right]^2$$

$$= \mathbb{E}\left[ U(\Theta)\langle \Theta, \Theta \rangle - \langle \hat{\Theta}, \Theta \rangle \right]^2 = \mathbb{E}\left[ \langle U(\Theta)\Theta, \Theta - \hat{\Theta} \rangle \right]^2.$$

*Finally, we apply Lemma 3 with $U = U(\Theta)\Theta$ and $V = \Theta - \hat{\Theta}$ and rearrange terms to obtain the claimed result:*

$$\mathbb{E}\left[ \langle U(\Theta)\Theta, \Theta - \hat{\Theta} \rangle \right]^2 \leq d \cdot \mathbb{E}\left[ \left( \langle U(\Theta)\Theta, \tilde{\Theta} - \hat{\Theta} \rangle \right)^2 \right]$$

$$= d \cdot \mathbb{E}\left[ U(\Theta)^2 \mathbb{E}\left[ \langle \Theta, \tilde{\Theta} - \hat{\Theta} \rangle^2 | \Theta \right] \right]$$

$$= d \cdot \mathbb{E}\left[ \frac{\mathbb{V}\left[ \varphi_\beta\left(1 - \langle \hat{\Theta}, \Theta \rangle\right) \mid \Theta \right]}{\mathbb{V}\left[ 1 - \langle \hat{\Theta}, \Theta \rangle \mid \Theta \right]} \mathbb{V}\left[ \langle \hat{\Theta}, \Theta \rangle \mid \Theta \right] \right]$$

$$= d \cdot \mathbb{E}\left[ \mathbb{V}\left[ \varphi_\beta\left(1 - \langle \hat{\Theta}, \Theta \rangle\right) \mid \Theta \right] \right].$$

Putting together Lemma 2 and Lemma 5, we get that the information ration $\Gamma(\Theta, \hat{\Theta})$ can be bounded by

$$\Gamma(\Theta, \hat{\Theta}) \leq d/2 \cdot \frac{\mathbb{E}\left[ \mathbb{V}\left[ \varphi_\beta\left(1 - \langle \hat{\Theta}, \Theta \rangle\right) \mid \Theta \right] \right]}{\mathbb{E}\left[ \mathbb{V}\left[ \psi_\beta\left(1 - \langle \hat{\Theta}, \Theta \rangle\right) \mid \Theta \right] \right]}.$$

The last part of the proof, Appendix B.3, takes care of controlling the ratio of expected variances over the functions $\varphi_\beta$ and $\psi_\beta$.

### B.3 Bounding the ratio of expected variances over the functions $\varphi_\beta$ and $\psi_\beta$

By definition, the function $\psi_\beta$ and its surrogate $\varphi_\beta$ are equal for $x \in [0, \delta_\beta]$ and then diverge linearly at a rate of $\psi_\beta(\delta_\beta)/\delta_\beta$. We observe, in Remark 1, that $\delta_\beta$ is a decreasing function of $\beta$ and that $\psi_\beta(\delta_\beta)/\delta_\beta$ strictly increases with $\beta$. This observation suggests that studying the case $\beta \to \infty$ could provide a general upper bound. Indeed, taking the limit case $\beta \to \infty$, the domain where the two functions differ is maximized, and the rate at which they differ is the largest. We show in Lemma 7

that under some simple transformations, increasing the value of $\beta$ does lead to a larger ratio of expected variances, and therefore, the case $\beta$ tending to $\infty$ can serve to derive general upper bounds. Quite satisfyingly, this limit case will provide a lot of simplifications. We will prove in Lemma 7, that the ratio of expected variance between $\psi_\beta$ and $\varphi_\beta$ can be upper bounded by the ratio of expected variance between $\overline{\psi}$ and $\overline{\varphi}$ defined as

$$\overline{\psi}(x) = \begin{cases} 0 & x \in [0,1], \\ 1 & x \in ]1,2], \end{cases} \tag{3}$$

and

$$\overline{\varphi}(x) = \begin{cases} 0 & x \in [0,1], \\ 1 + 2(x-1) & x \in ]1,2]. \end{cases} \tag{4}$$

We present first the analysis of the ratio of expected variance for the case $\beta$ tending to $\infty$ and then justify studying the case $\beta \to \infty$ as an upper bound to the general case in Lemma 7.

**Lemma 6** *Let $\overline{\psi}$ and $\overline{\varphi}$ be defined respectively in (3) and (4). Then, it holds that*

$$\frac{\mathbb{E}\left[\mathbb{V}\left[\overline{\varphi}\left(1-\langle\hat{\Theta},\Theta\rangle\right) \mid \Theta\right]\right]}{\mathbb{E}\left[\mathbb{V}\left[\overline{\psi}\left(1-\langle\hat{\Theta},\Theta\rangle\right) \mid \Theta\right]\right]} \leq 9.$$

**Proof 5** *We start by analyzing $\mathbb{E}\left[\mathbb{V}\left[\overline{\psi}\left(1-\langle\hat{\Theta},\Theta\rangle\right) \mid \Theta\right]\right]$. We note that $\overline{\psi}\left(1-\langle\hat{\Theta},\Theta\rangle\right)$ is equal to 1 if $\langle\hat{\Theta},\Theta\rangle < 0$ and is equal to 0 otherwise. To distinguish those two cases, we introduce the notation $I(\hat{\Theta},\Theta) := \mathbb{1}_{\{\langle\hat{\Theta},\Theta\rangle<0\}}$. We observe that $\mathbb{E}\left[\mathbb{V}\left[\overline{\psi}\left(1-\langle\hat{\Theta},\Theta\rangle\right) \mid \Theta\right]\right]$ is equal to the expected variance of Bernoulli random variable with probability given by $Q(\Theta) := \mathbb{E}[I(\hat{\Theta},\Theta)]$ and can therefore be written as*

$$\mathbb{E}\left[\mathbb{V}\left[\overline{\psi}\left(1-\langle\hat{\Theta},\Theta\rangle\right) \mid \Theta\right]\right] = \mathbb{E}[Q(\Theta)(1-Q(\Theta))].$$

*The last part of the proof concerns $\mathbb{E}\left[\mathbb{V}\left[\overline{\varphi}\left(1-\langle\hat{\Theta},\Theta\rangle\right) \mid \Theta\right]\right]$. Similarly, we can distinguish between two cases: either $\langle\hat{\Theta},\Theta\rangle \geq 0$ and $\overline{\varphi}(1-\langle\hat{\Theta},\Theta\rangle) = 0$, or $\langle\hat{\Theta},\Theta\rangle < 0$ and $\overline{\varphi}(1-\langle\hat{\Theta},\Theta\rangle) = 1 - 2\langle\hat{\Theta},\Theta\rangle$. Introducing the notation $G(\Theta) := \mathbb{E}[I(\hat{\Theta},\Theta)\langle\hat{\Theta},\Theta\rangle]$, we can write*

$$\mathbb{E}\left[\mathbb{V}\left[\overline{\varphi}\left(1-\langle\hat{\Theta},\Theta\rangle\right) \mid \Theta\right]\right] = \mathbb{E}\left[\mathbb{E}\left[\left(\overline{\varphi}\left(1-\langle\hat{\Theta},\Theta\rangle\right) - \mathbb{E}\left[\overline{\varphi}\left(1-\langle\hat{\Theta},\Theta\rangle\right) \mid \Theta\right]\right)^2 \mid \Theta\right]\right]$$

$$= \mathbb{E}\left[I(\hat{\Theta},\Theta)\left(1 - 2\langle\hat{\Theta},\Theta\rangle - (Q(\Theta) + 2G(\Theta))\right)^2\right]$$

$$+ \mathbb{E}\left[\left(1 - I(\hat{\Theta},\Theta)\right)(0 - (Q(\Theta) + 2G(\Theta)))^2\right].$$

*Distributing the square and simplifying terms, we obtain*

$$\mathbb{E}\left[I(\hat{\Theta},\Theta)\left(1 - 2\langle\hat{\Theta},\Theta\rangle\right)^2\right] - 2\mathbb{E}\left[I(\hat{\Theta},\Theta)\left(1 - 2\langle\hat{\Theta},\Theta\rangle\right)(Q(\Theta) + 2G(\Theta))\right]$$

$$+ \mathbb{E}\left[I(\hat{\Theta},\Theta)(Q(\Theta) + 2G(\Theta))^2\right] + \mathbb{E}\left[\left(1 - I(\hat{\Theta},\Theta)\right)(Q(\Theta) + 2G(\Theta))^2\right]$$

$$= \mathbb{E}\left[I(\hat{\Theta},\Theta)\left(1 - 2\langle\hat{\Theta},\Theta\rangle\right)^2\right] - \mathbb{E}\left[(Q(\Theta) + 2G(\Theta))^2\right].$$

*To get to the last part of the proof, we rewrite explicitly $Q(\Theta) + 2G(\Theta)$ as $\mathbb{E}\left[I(\hat{\Theta},\Theta)\left(1 - 2\langle\hat{\Theta},\Theta\rangle\right)\right]$ and optimize over the values of $(1 - 2\langle\hat{\Theta},\Theta\rangle)$. We can that*

$$\mathbb{E}\left[I(\hat{\Theta},\Theta)\left(1 - 2\langle\hat{\Theta},\Theta\rangle\right)^2\right] - \mathbb{E}\left[\mathbb{E}\left[I(\hat{\Theta},\Theta)\left(1 - 2\langle\hat{\Theta},\Theta\rangle\right)\right]^2\right]$$

$$\leq \sup_{\alpha\in[-1,3]} \mathbb{E}\left[I(\hat{\Theta},\Theta)\alpha^2\right] - \mathbb{E}\left[\mathbb{E}\left[I(\hat{\Theta},\Theta)\alpha\right]^2\right] = 9 \cdot \mathbb{E}[Q(\Theta)(1 - Q(\Theta))],$$

*which concludes the proof.*

**Lemma 7** *Let $\psi_\beta(x) = \phi_\beta(1) - \phi(1-x)$ and the logistic surrogate $\varphi_\beta$ as in Definition 2 and let $\overline{\psi}$ and $\overline{\varphi}$ be defined respectively in (3) and (4). Then, for all $\beta > 0$, it holds that*

$$\frac{\mathbb{E}\left[\mathbb{V}\left[\varphi_\beta\left(1 - \langle\hat{\Theta}, \Theta\rangle\right) | \Theta\right]\right]}{\mathbb{E}\left[\mathbb{V}\left[\psi_\beta\left(1 - \langle\hat{\Theta}, \Theta\rangle\right) | \Theta\right]\right]} \leq \frac{\mathbb{E}\left[\mathbb{V}\left[\overline{\varphi}\left(1 - \langle\hat{\Theta}, \Theta\rangle\right) | \Theta\right]\right]}{\mathbb{E}\left[\mathbb{V}\left[\overline{\psi}\left(1 - \langle\hat{\Theta}, \Theta\rangle\right) | \Theta\right]\right]}.$$

**Proof 6** *Beginning with the ratio of expected variances between $\varphi_\beta$ and $\psi_\beta$, we will apply a series of transformations to the functions $\varphi_\beta$ and $\psi_\beta$, ultimately yielding the functions $\overline{\varphi}$ and $\overline{\psi}$. These transformations are chosen to ensure that they can only increase the ratio of expected variances.*

*By definition, the function $\psi_\beta$ and its surrogate $\varphi_\beta$ are identical for $x \in [0, \delta_\beta)$ and then diverge linearly at a rate of $\psi_\beta(\delta_\beta)/\delta_\beta$ on the interval $x \in [\delta_\beta, 2]$. We illustrate this on Figure 1. Focusing on the domain where the two functions coincide, we observe that the transformation $f(x) = \max(x, \psi_\beta(1))$ reduces the expected variance for both $\psi_\beta$ and $\varphi_\beta$. However, since $\psi_\beta(x)$ is less than or equal to $\varphi_\beta(x)$ for all $x \in [0, 2]$, and both functions exceed $\psi_\beta(1)$ on the interval $[1, 2]$, the transformation $f$ proportionally reduces the expected variance of $\psi_\beta$ more than that of $\varphi_\beta$. As a result, the transformation increases the ratio of expected variances between the two functions. As $\psi_\beta$ and $\varphi_\beta$ are strictly increasing functions, the resulting functions, illustrated on Figure 2, can be written as*

$$f(\psi_\beta(x)) = \begin{cases} \psi_\beta(1) & x \in [0, 1], \\ \psi_\beta(x) & x \in ]1, 2] \end{cases}$$

*and*

$$f(\varphi_\beta(x)) = \begin{cases} \psi_\beta(1) & x \in [0, 1], \\ \varphi_\beta(x) & x \in ]1, 2]. \end{cases}$$

*The second transformation we apply concerns only the function $f(\psi_\beta(x))$. We will crop all the values larger than $\psi(\delta_\beta)$ by applying the transformation $g(x) = \min(x, \psi(\delta_\beta))$. As $f(\psi_\beta(x))$ is an increasing function, the function $g(f(\psi_\beta(x)))$, illustrated on Figure 3, can be written as*

$$g(f(\psi_\beta(x))) = \begin{cases} \psi_\beta(1) & x \in [0, 1], \\ \psi_\beta(x) & x \in ]1, \delta_\beta] \\ \psi_\beta(\delta_\beta) & x \in ]\delta_\beta, 2]. \end{cases}$$

*The transformation $g$ reduces the variance of the function $f(\psi_\beta(x))$ as it both decreases the values of $f(\psi_\beta(x))$ and the derivative of $f(\psi_\beta(x))$ for all $x \in ]\delta_\beta, 2]$.*

*The third transformation we apply is increasing the value of $\beta$. As $\beta$ increases, the derivative of $f(\varphi_\beta(x))$ increases everywhere,*

$$\frac{d}{dx} f(\varphi_\beta(x)) = \begin{cases} 0 & x \in [0, 1], \\ \frac{\beta \exp(-\beta(1-x))}{(1+\exp(-\beta(1-x)))^2} & x \in ]1, \delta_\beta] \\ \psi_\beta(\delta_\beta)/\delta_\beta & x \in ]\delta_\beta, 2], \end{cases}$$

*and the expected variance of $f(\varphi_\beta)$ increases. Regarding $g(f(\psi_\beta(x)))$, we can show that that for all $x \in [0, 2]$, the ratio $f(\varphi_\beta(x))/g(f(\psi_\beta(x)))$ increases with $\beta$. Indeed, this ratio is equal to $1$ for all $x \in [0, \delta_\beta]$ and increases for all $x \in ]\delta_\beta, 2]$ as*

$$\frac{f(\varphi_\beta(x))}{g(f(\psi_\beta(x)))} = \frac{\varphi_\beta(\delta_\beta) + \varphi_\beta(\delta_\beta)/\delta_\beta \cdot (x - \delta_\beta)}{\varphi_\beta(\delta_\beta)} = 1 + \frac{(x - \delta_\beta)}{\delta_\beta}$$

*and as $\delta_\beta$ is a decreasing function of $\beta$ (see Remark 1), the ratio $(x - \delta_\beta)/\delta_\beta$ is a increasing function of $\beta$ for all $x \in ]\delta_\beta, 2]$. This fact ensures that the expected variance of $g(f(\psi_\beta(x)))$ cannot increase proportionally more than the expected variance of $f(\varphi_\beta(x))$. We can therefore study the ratio of expected variances between $f(\varphi_\infty)$ and $g(f(\psi_\infty))$.*

*The last operation we apply is merely a convenient shifting and scaling, $h(x) = (x - g(f(\psi_\beta(1))))/(g(f(\psi_\infty(2))) - g(f(\psi_\beta(1))))$. Applied on both $g(f(\psi_\infty))$ and $f(\varphi_\infty)$ these operations do not affect the ratio of expected variances. The resulting functions are illustrated on Figure 4.*

*To express the resulting functions, we have to analyze the function $\psi_\beta(x)$ for $\beta$ tending to infinity for values $x \in ]1, 2]$.*

*We recall that $\psi_\beta(x) = \phi_\beta(1) - \phi_\beta(1 - x)$ and can equivalently be written as*

$$\psi_\beta(x) = \frac{1}{1 + \exp(-\beta)} - \frac{1}{1 + \exp(-\beta(1 - x))}.$$

*We have to distinguish between three cases for $(x - 1)$: negative, zero, or positive. For values of $x \in ]1, 2]$, we have that $(1 - x) < 0$ and that $\lim_{\beta \to \infty} \psi_\beta(x) = 1$, if $x = 1$, we have that $\lim_{\beta \to \infty} \psi_\beta(x) = 1/2$ and for values of $x \in [0, 1[$, we have that $(1 - x) > 0$ and that $\lim_{\beta \to \infty} \psi_\beta(x) = 0$. We can then write*

$$\psi_\infty(x) = \begin{cases} 0 & x \in [0, 1[, \\ 1/2 & x = 1, \\ 1 & x \in ]1, 2]. \end{cases}$$

*We can now construct the corresponding $\varphi_\infty(x)$. We note that $\frac{\psi_\infty(x)}{x}$ is maximized when taking the limit to $x = 1^+$ from the right: $\lim_{x \to 1^+} \frac{\psi_\infty(x)}{x} = 1$. It comes that $\varphi_\infty(x)$ can be written as*

$$\varphi_\infty(x) = \begin{cases} 0 & x \in [0, 1[ \\ 1/2 & x = 1, \\ 1 + (x - 1) & x \in ]1, 2]. \end{cases}$$

*We denote the resulting functions $h(g(f(\psi_\infty(x))))$ and $h(f(\varphi_\infty(x)))$ respectively as $\overline{\psi}$ and $\overline{\varphi}$. We note that they can be written quite simply as*

$$\overline{\psi}(x) = \begin{cases} 0 & x \in [0, 1], \\ 1 & x \in ]1, 2], \end{cases}$$

*and*

$$\overline{\varphi}(x) = \begin{cases} 0 & x \in [0, 1], \\ 1 + 2(x - 1) & x \in ]1, 2]. \end{cases}$$

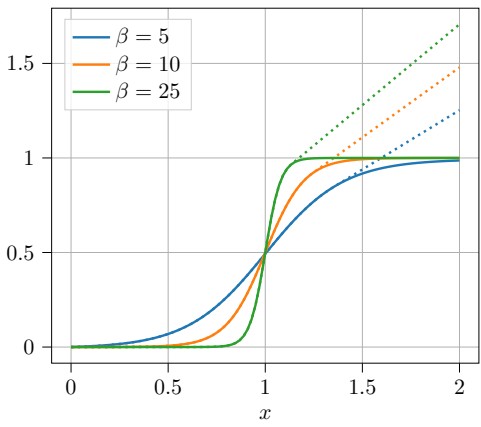

Figure 1: Illustration of the function $\psi_\beta$ (in solid line) and the function $\varphi_\beta$ (in dotted line) for different values of $\beta$.

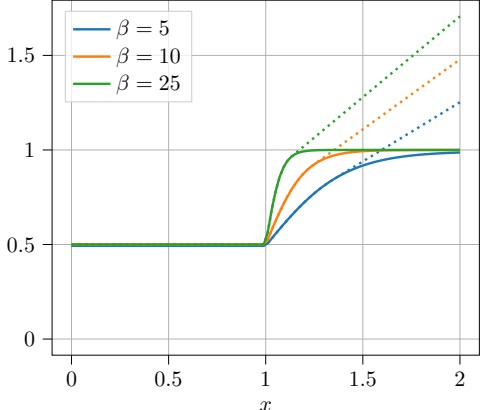

Figure 2: Illustration of the function $f(\psi_\beta)$ (in solid line) and the function $f(\varphi_\beta)$ (in dotted line) for different values of $\beta$.

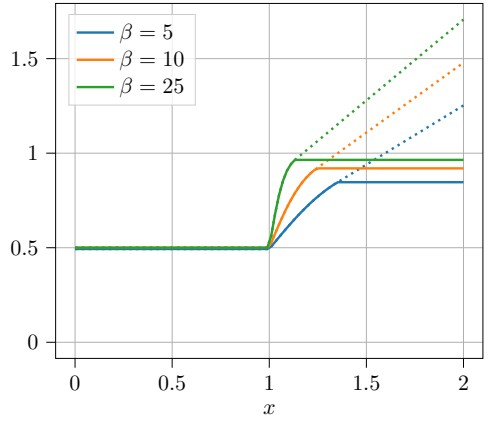

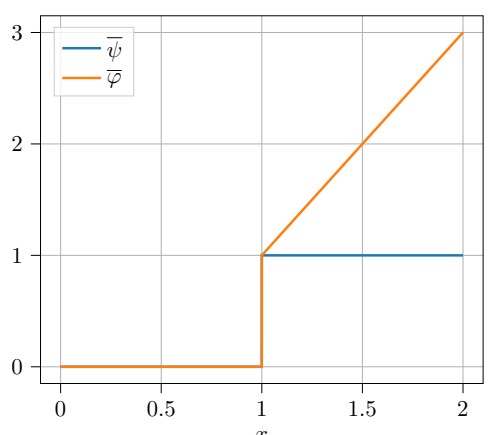

Figure 3: Illustration of the function $g(f(\psi_\beta))$ (in solid line) and the function $f(\varphi_\beta)$ (in dotted line) for different values of $\beta$.

Figure 4: Illustration of the function $\overline{\psi}$ (in blue) and the function $\overline{\varphi}$ (in orange).

**Remark 1** *We illustrate the function $\psi_\beta(x)/x$ on Figure 5 and the behavior of $\delta_\beta$ and $\psi_\beta(\delta_\beta)/\delta_\beta$ for increasing values of $\beta$ on Figure 6. The derivative of the function $\psi_\beta(x)/x$ is given by*

$$\frac{d}{dx}\left(\frac{\psi_\beta(x)}{x}\right) = \frac{1}{x}\left(\frac{d}{dx}\psi_\beta(x) - \frac{\psi_\beta(x)}{x}\right).$$

*We note that it is equal to zero for values of $x \in ]0,2]$ such that $\frac{d}{dx}\psi_\beta(x) = \frac{\psi_\beta(x)}{x}$. By definition of $\delta_\beta$, we have $\frac{d}{dx}\psi_\beta(\delta_\beta) = \frac{\psi_\beta(\delta_\beta)}{\delta_\beta}$.*

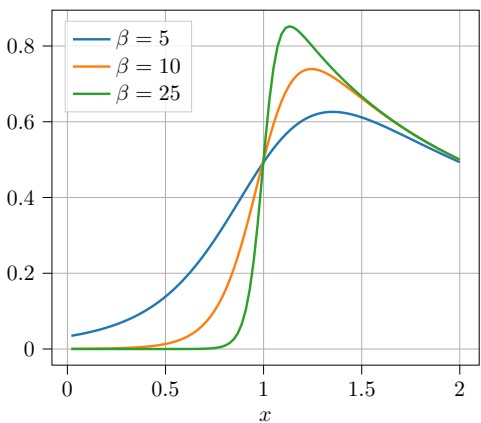

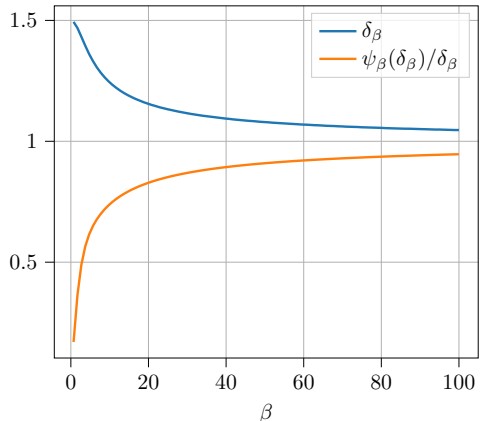

Figure 5: Illustration of the function $\psi_\beta(x)/x$ for different values of $x$. The maximum of the function is attained for $x = \delta_\beta$.

Figure 6: Illustration of $\delta_\beta$ and $\psi_\beta(\delta_\beta)/\delta_\beta$ as functions of $\beta$. One can observe that $\delta_\beta$ decreases with $\beta$ while $\psi_\beta(\delta_\beta)/\delta_\beta$ increases.

## C  Proof of Theorem 2

This section presents the proof of Theorem 2. One lemma, (van Handel, 2016, Lemma 5.13), will be particularly useful to control the covering number. We restate the lemma below.

**Lemma 8 ((van Handel, 2016, Lemma 5.13))** *Let $\mathbf{B}_d(0,1)$ denote the $d$-dimensional closed Euclidean unit ball. We have $|\mathcal{N}(\mathbf{B}_d(0,1), ||\cdot||_2, \varepsilon)| = 1$ for $\varepsilon \geq 1$ and for $0 < \varepsilon < 1$, we have*

$$\left(\frac{1}{\varepsilon}\right)^d \leq |\mathcal{N}(\mathbf{B}_d(0,1), ||\cdot||_2, \varepsilon)| \leq \left(1 + \frac{2}{\varepsilon}\right)^d.$$

**Theorem 2** *Under the logistic bandit setting with logistic function $\phi_\beta(x) = e^{\beta x}/(1 + e^{\beta x})$, let $\mathcal{A} \subseteq \mathbf{B}_d(0,1)$ and $\mathcal{O} \subseteq \mathbf{S}_d(0,1)$ be such that $\mathcal{O} \subseteq \mathcal{A}$. Then for all $\beta > 0$, the TS regret is bounded as*

$$\mathbb{E}[\text{Regret}(T)] \leq 3d\sqrt{T\log\left(\sqrt{3 + \frac{6\beta T}{d}}\right)}.$$

**Proof 7** *The proof starts by combining Theorem 1 with Proposition 1. We can then write*

$$\mathbb{E}[\text{Regret}(T)] \leq 3\sqrt{dT/2\left(\mathrm{H}(\Theta_\varepsilon) + \varepsilon\beta T\right)},$$

*where $\Theta_\varepsilon$ is defined for some $\varepsilon > 0$ as in Definition 1. To define $\Theta_\varepsilon$, we can set $\mathcal{O}_\varepsilon$ as the $\varepsilon$-net of smallest cardinality. The entropy $\mathrm{H}(\Theta_\varepsilon)$ is upper bounded by the logarithm of the cardinality of the space $\mathcal{O}_\varepsilon$ (see (Yury Polyanskiy, 2022, Theorem 1.4.b)), which corresponds to the logarithm of the covering number $\mathcal{N}(\mathcal{O}, \rho, \varepsilon)$. As $\mathcal{O} \subseteq \mathbf{B}_d(0,1)$ and $\rho$ is the Euclidean distance, we can apply Lemma 8 and upper bound the TS regret as*

$$\mathbb{E}[\text{Regret}(T)] \leq 3\sqrt{dT/2\left(d\log\left(1 + \frac{2}{\varepsilon}\right) + \varepsilon\beta T\right)}.$$

*As the bound holds for any $\varepsilon > 0$, we can set $\varepsilon = d/(\beta T)$. Using properties of the logarithm, we arrive at the claimed inequality:*

$$\mathbb{E}[\text{Regret}(T)] \leq 3d\sqrt{T/2\left(\log\left(1 + \frac{2\beta T}{d}\right) + 1\right)}$$

$$\leq 3d\sqrt{T/2\log\left(3 + \frac{6\beta T}{d}\right)} = 3d\sqrt{T\log\left(\sqrt{3 + \frac{6\beta T}{d}}\right)}.$$

