# OpenReview forum: "An Information-Theoretic Analysis of Thompson Sampling for Logistic Bandits"
_NeurIPS.cc/2024/Workshop/BDU — NeurIPS BDU Workshop 2024 Poster_

### Official Review · Reviewer_hnvv · 2024-09-22
**A Rigorous Theoretical Analysis of Thompson Sampling for Logistic Bandits, But Lacking Practical Validation**

**Rating:** 6
**Confidence:** 4

**Review:**

The authors present a well-structured analysis of Thompson Sampling algorithm for logistic bandits. It focuses on a refined information-theoretical approach which is built upon previous works relevant to Thompson Sampling by improving regret bounds. Compared to previous works, it aims to address the exponential dependence on the slope parameter with the mathematical proof. The proposed regret bounds provide a better theoretical understanding of Thompson Sampling for logistic bandits.

However, although the pre-defined problem, algorithms, and theoretical derivations are explained step by step, the current version lacks convincing results based on numerical demonstrations. To broaden the readership, the authors are strongly suggested to devise numerical simulation examples to demonstrate their claims, as well as providing simulation scripts for the numerical examples with clear step-by-step comments. Otherwise, this kind of mathematical derivation is hardly to be useful for the broad readership of the booming community.

In a word, the current version contribution to the community would be incremental due to its complexity and limited numerical validation, which could hamper for a broader readership or practical implementation for such theoretical proposal. The authors are expected to revise accordingly.

---

### Official Review · Reviewer_UaYR · 2024-09-26

**Rating:** 6
**Confidence:** 3

**Review:**

Summary:
----------
The Thompson algorithm for logistic bandits is studied under the assumption that the parameter space is included inside the action space.  A new bound on the information ratio is derived, as well as a regret bound which scales much better than previous results and is valid for a continuous action space.

Main Review:
----------
Quality and clarity: Generally, the paper is well-written. However, I think that switching from $\theta$ to $\Theta$ causes confusion (if I understand correctly they denote the same object, the unknown model parameter).

I do not understand the last equality before l. 315. Shouldn't it be
$$
U(\Theta) \langle \Theta, \Theta \rangle - \langle \hat{\Theta}, \Theta \rangle
= \langle U(\Theta) \Theta - \hat{\Theta}, \Theta \rangle
$$
instead of $\langle U(\Theta) \Theta, \Theta - \hat{\Theta} \rangle$?


Typos
--------
Proof 1: $\sum_{t = 1}^{T}$ missing before 221.
Proof 2: In the formula after line 276, the inequality is written the wrong way around.
Lemma 8: missing | after the covering number for $\epsilon \ge 1$.
Proof 7: in the last inequality, it should be $2 \beta T / d$ instead of $2 \beta T / \beta$.
Remark 1: full stop missing after the formula after l. 392. Left-over words in l. 394.
The $(x)$ in ll. 331, 335, 349 are superfluous.
l. 361: "functions" instead of "function".
l. 366: should be instead (I think) "$f(\psi_{\beta}(x))$ as it".

Citations
--------
Citation "Information Theory - From Coding to Learning" outdated: one author missing, correct year should be 2024.
For Russo, Van Roy - "An Information-Theoretic Analysis of Thompson Sampling", please cite the version published in JMLR.
Similarly, Russo, Van Roy - "Learning to Optimize via Informationen-Directed Sampling" has been published in NeurIPS, Faury et. al's paper in PMLR, Dong and Van Roy's paper in NIPS'18 etc. (I didn't do through the other arXiv citations).

Minor mathematical stuff
---------
Proof 2: first you define $h_2$ on (0, 1), but then plug in values in [0, 1] into it in line 275. One can extend $h_2$ by $0$ onto $[0, 1]$, however and consider one-sided derivatives at the endpoint, but right now you are mixing both concepts.
In line 273 it should instead say "$\xi$ in between $x$ and $p$". Since $x$ could be larger than $p$, and then $(x, p)$ is empty.
In line 276, I recommend instead writing "for $x = U(\omega)$ and $p = \mathbb{E}[U]$ for $\omega \in \Omega$" to be more precise. Then the inequality holds for all $\omega \in \Omega$ and taking the expectation then yields the next formula, as desired.
Also, according to my experience, the Shannon entropy is defined with respect to $\log_2$ instead of $\ln$, and in that case, your derivative of $h_2$ is of by a multiplicative factor (which hence does not impede the proof at all).


Other comments / wording
--------
In line 52, I would write "scales only logarithmically with" instead of "on".
In line 10, I would write "regret bound of" instead of "in" (agreeing with other similar formulations in the rest).
Why do you use http:// instead of https:// for the arXiv links?
Also the formatting (e.g. when italics are used) is not entirely consistent, compare for example lines 202-203 with 205-206.

---

### Decision · Program_Chairs · 2024-10-09

Accept (Poster)